# Platelet count/spleen volume ratio has a good predictive value for esophageal varices in patients with hepatitis B liver cirrhosis

**Sihao Yu**[1‡], **Wei Chen**[1]*, **Zicheng Jiang**[2]*

1 Department of Gastroenterology, Xiangyang No.1 People's Hospital, Hubei University of Medicine, Xiangyang City, Hubei Province, P.R.C, 2 Department of Infectious Diseases, Ankang Central Hospital, Hubei University of Medicine, Ankang City, Shanxi Province, P.R.C

☯ These authors contributed equally to this work.
‡ These author share first authorship on this work.
* chenweiyf@163.com (WC); jzc1977@126.com (ZJ)

**Data Availability Statement:** All relevant data are within the manuscript and its Supporting Information files.

## Abstract

### Background & aims

Platelet count/spleen longest diameter ratio (PSDR) is widely used in clinical practice due to its good performance in predicting esophageal varices (EV). We obtained spleen volume (SV) by magnetic resonance examination, the purpose of this study was to evaluate the clinical value of platelet count/spleen volume ratio (PSVR) and spleen volume in predicting EV in patients with hepatitis B cirrhosis. Methods: This study was a diagnostic accuracy experiment and retrospective, 199 patients with hepatitis B cirrhosis who met the criteria were selected as the research subjects. All patients were collected blood samples in the morning on an empty stomach within 2 days, and related indicators were tested. Within 10 days, they received electronic gastroscopy and abdominal magnetic resonance examination. According to the Child-Pugh score, the patients were divided into groups with or without EV and with or without high-risk esophageal varices (HRV), then statistical analysis of the two groups was performed.

### Results

The area under the curve (AUC) of PSVR in predicting EV or HRV in each group (85.5%-92.6%) was higher than PSDR, SV, spleen diameter, and platelet count. The AUC of PSDR in diagnosing HRV was higher than SV, and the AUC of SV in diagnosing EV was higher than PSDR, but the difference was not significant (P>0.05). In Child-Pugh A patients, Multivariate logistic regression analysis showed PSVR could be a predictor of HRV (P<0.05), SV was a reliable predictor of EV (P<0.05).

### Conclusion

PSVR is better than PSDR, spleen diameter, platelet count in predicting EV; in the absence of serological results, SV could be used instead of PSDR. Both can predict EV or HRV of patients with hepatitis B cirrhosis.

**Funding:** The author(s) received no specific funding for this work.

**Competing interests:** The authors have declared that no competing interests exist.

## Introduction

Cirrhosis is a chronic disease of the liver, which is often associated with many severe complications. In particular, gastroesophageal varices (GEV) is a complication characterized by swollen or enlarged veins and develops at a rate of about 7–8% per year. The rate of variceal development from small to large varices is about 10–12% per year, while variceal bleeding occurs at a rate of 5% per year, with a 15–25% mortality at six weeks [1, 2]. Clinical guidelines recommend one diagnostic gastroscopy procedure every two to three years for patients without varices. Conversely, guidelines recommend repeated gastroscopy procedures every one to two years in patients with small varices [3, 4]. However, gastroscopy is an invasive and relatively expensive procedure accompanied by numerous related risks.

Currently, there are many diagnostic procedures available for the non-invasive prediction of GEV, which have good accuracy in diagnosing esophageal varices. These tests include the liver stiffness measurement (LSM), spleen stiffness measurement, platelet count/spleen longest diameter ratio (PSDR). Moreover, the Baveno VI consensus suggests forgoing endoscopic screening for patients whose liver stiffness values are less than 20 kPa, accompanied by platelet (PLT) counts over $150 \times 10^9$/L. This exclusion is due to the relatively small risk (less than 5%) of varices that require treatment [5, 6], Although Kazemi [7] confirmed LSM's superior performance compared to PSDR in predicting the appearance of more severe esophageal varices (EVs) at the area under the curve (AUC) 0.83 vs. 0.80; however, the difference is not statistically significant.

Simultaneously, LSM has not shown better performance than spleen diameter (SD) and platelet count in predicting EVs [8]. Recently, Hou J et al. [9] demonstrated good performance (sensitivity of 95.7%, specificity of 76.4%, negative predictive value of 98.57%, negative likelihood ratio of 0.06) of the Baveno VI consensus when combined with the spleen stiffness measurement ($\leq$46 kPa), in the identification of high-risk esophageal varices (HRV, defined as medium to large varices or small varices with red color signs according to the current Baveno VI consensus) in patients with HBV-related cirrhosis of maintained viral suppression. However, in patients with obesity, narrow intercostal space, and ascites (abnormal abdominal fluid buildup), the failure rate of the liver and spleen stiffness measurement is as high as 10.7% [8]. Therefore, its application is relatively limited in clinical practice. In recent years, as people's economic conditions have improved, abdominal magnetic resonance imaging (MRI) has become more affordable and more widely accepted for the measurement of spleen volume (SV). This study intends to explore the performance of SV and platelet count/spleen volume ratio (PSVR), examine its relationships with EV and HRV to improve clinical evaluations and treatment regimens.

## Materials and methods

### Ethics statement

According to the Declaration of Helsinki (as revised in Brazil 2013), this study was approved by the Ethics Committee of Xiangyang No. 1 People' s Hospital Hubei University of Medicine (approval no. 2020KY043). Since the data were obtained retrospectively from the hospital's database, there was no need to sign an informed consent form.

The study was completed in December 2020. Patients with hepatitis B and cirrhosis diagnosed from May 2016 to September 2020 in the Department of Gastroenterology, Xiangyang No.1 People's Hospital, Hubei University of Medicine were selected as the research objects. Fluorescence quantitative polymerase chain reaction was used to diagnose hepatitis B virus infection, and liver cirrhosis was diagnosed by histological examination or combined with

physical examination, laboratory, and radiological examination. Exclusion criteria: 1) had a family history of platelet count reduction; 2) had a history of hemophilia; 3) had a history of liver malignancies; 4) had recently taken immunosuppressants or other drugs that may affect platelet count; 5) had a history of liver-related surgery or local treatment; 6) after splenectomy; 7) gastrointestinal bleeding and other clinical manifestations of bleeding within 3 months; 8) cases of splenomegaly caused by blood diseases were removed; 9) after GEV's injection of the sclerosing agent or band ligation; 10) cirrhosis caused by factors other than hepatitis B virus infection; 11) had serious complications such as hepatic encephalopathy, hepatorenal syndrome, etc; 12) had other diseases that may affect the test results (chronic obstructive pulmonary disease, asthma, aortic stenosis, and so on). The authors were not exposed to information that could identify individual participants during or after data collection.

## Serological indicators

All patients received blood sampling in the early morning on an empty stomach within 2 days after admission (hospitalized for abnormal liver function or symptoms related to liver diseases such as ascites and jaundice) to test routine blood, liver and kidney function, electrolytes, coagulation function, five indicators of liver fibrosis, and five indicators of hepatitis B virus. Gastroscopy and abdominal magnetic resonance examination were performed by our senior physicians within 10 days after admission to the hospital.

## MRI

The spleen volume was calculated using the MRI technique, comparing the spleen as a long ellipsoid, using the standard long ellipsoid formula: $0.523 \times W \times T \times L$ (W = width, T = thickness, L = length) (Fig 1). The extent of ascites was obtained by examination findings, MRI, and color doppler ultrasound (if available): Grade 1 or mild ascites, which can only be detected by imaging such as MRI or ultrasound; grade 2 or moderate ascites, which presents with moderate symmetric abdominal distention; and grade 3 ascites, which is large or gross ascites with significant abdominal distention [10].

## Gastroscopy

According to the criteria proposed by the Baveno VI Consensus Conference [6], high-risk varices were considered if any of the following endoscopic features were met: (I) beaded or tumor-like varices; (II) varices with red color signs; (III) varices with blood clots; or (IV) the maximal diameter of varices > 5 mm.

The above results were obtained by staff with more than 5 years of work experience.

## Statistical analysis

SPSS 26.0 statistical software was applied for data analysis. Age, gender, smoking history, drinking history, mean arterial pressure, ascites, Child-Pugh (CP) grade, blood routine, liver and kidney function, electrolytes, coagulation function, magnetic resonance-related indexes, and relevant continuous variables in the elicited parameters were expressed as median and 25th and 75th percentiles, and categorical variables were expressed as frequencies and percentages. The Kruskal-Wallis H test was used for quantitative data, and the chi-square test was used for category data; $P < 0.05$ was considered a statistically significant difference.

Receiver operating characteristic (ROC) curves were plotted for PSVR, PSDR, SV, SD, platelet count, and cutoff values were determined using Youden index, and each data was compared with each group of esophageal varices detected by gastroscopy for performance

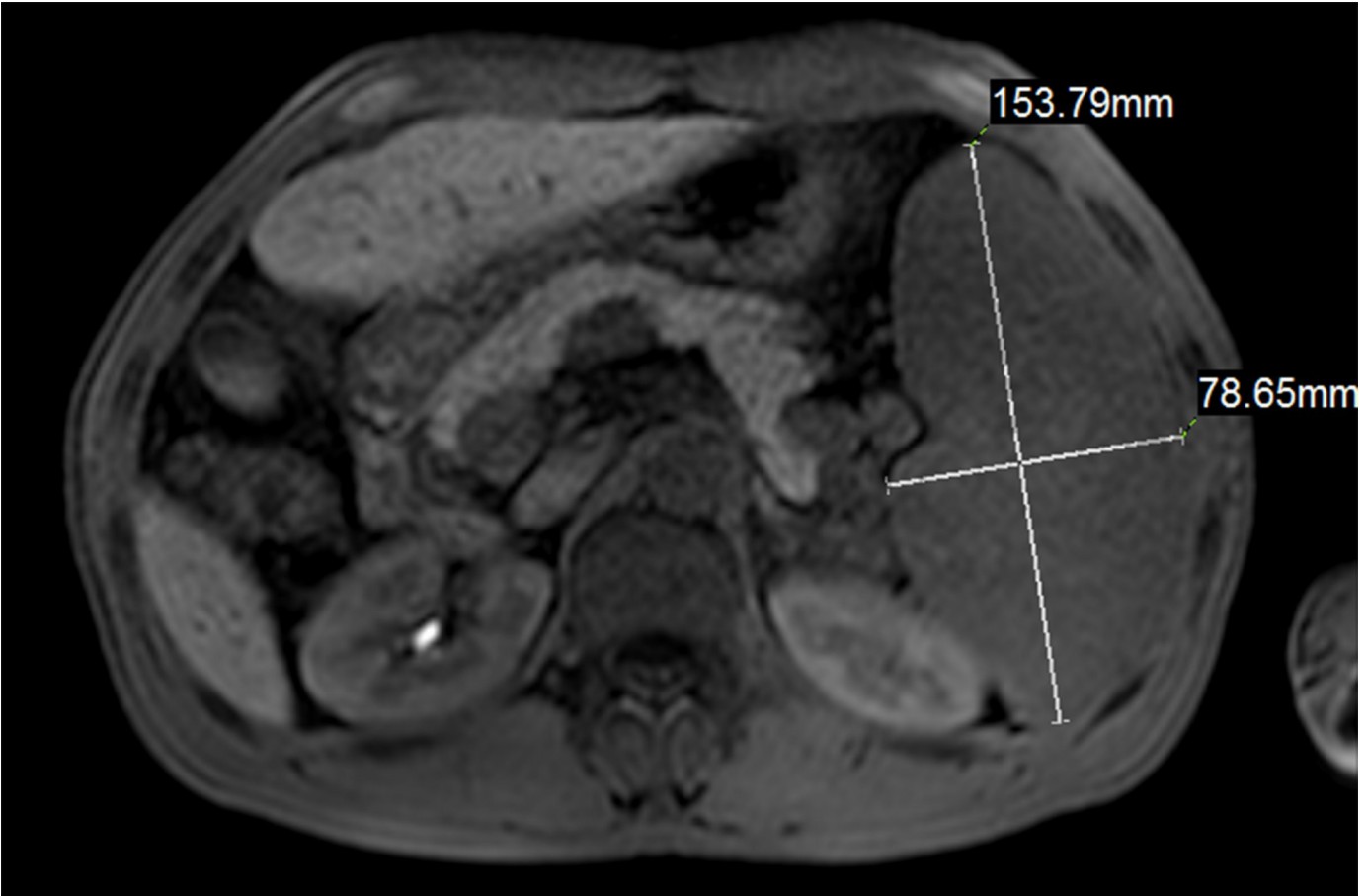

**Fig 1. MRI measurement of spleen width and thickness.**

evaluation, yielding accuracy, sensitivity, specificity, positive and negative predictive values. A collinearity test was performed, variables were excluded, and the Hosmer-Lemeshow test was performed followed by a binary logistic regression model to calculate P-value, coefficient of regression, odds ratio (OR), and 95% confidence interval (CI) for determining whether they were predictors.

## Result

The basic situation of the population was shown in Table 1. During the study period, 199 patients with hepatitis B liver cirrhosis met the inclusion criteria, all of them are Han people from Hubei Province and 147 of them entered the Child-Pugh A group. All patients were positive for hepatitis B virus surface antigen. Among the 199 cirrhosis patients studied, the median age was 52 (44–60) years, 127 (63.8%) were male, 64 (32.2%) had a history of smoking, 55 (27.6%) had a history of drinking. Among the 55 patients, the average daily drinking volume of males was less than 40 g/d, that of females was less than 20 g/d, 46 had a drinking history of fewer than 5 years, and 9 had a drinking history of more than 5 years, all of them had quit drinking. 18 (9.0%) had a history of diabetes, and 32 (16.1%) had a history of hypertension. 147 were Child-Pugh class A, 49 were class B and 3 were class C. 81 (40.7%) of 199 patients had EVs, 35 (17.6%) of patients had HRVs. In the Child-Pugh A group, the number of true negatives was 99, and it can be concluded that PSVR or SV could predict the absence of esophageal varices in 67%

**Table 1. Basic characteristics of the population.**

| | EV (-), n = 118 | EV (+), n = 81 | HRV, n = 35 |
|---|---|---|---|
| Age (years) | 52 (44–64) | 53 (46–58.5) | 51 (45–58) |
| Male | 71 (60.2) | 56 (69.1) | 24 (68.6) |
| Smoking history | 39 (33.1) | 25 (30.9) | 10 (28.6) |
| Drinking history | 31 (26.3) | 24 (29.6) | 10 (28.6) |
| Type 2 diabetes | 10 (8.5) | 8 (9.9) | 3 (8.6) |
| Arterial hypertension | 17 (14.4) | 15 (18.5) | 7 (20) |
| Ascites | | | |
| No | 96 (81.4) | 33 (40.7) | 10 (28.6) |
| Grade 1 | 19 (16.1) | 30 (37) | 12 (34.3) |
| Grade 2 | 3 (2.5) | 14 (17.3) | 9 (25.7) |
| Grade 3 | 0 | 4 (4.9) | 4 (11.4) |
| Child Pugh Score | | | |
| A | 99 (83.9) | 48 (59.3) | 21 (60) |
| B | 19 (16.1) | 30 (37) | 13 (37.1) |
| C | 0 | 3 (3.7) | 1 (2.9) |
| Laboratory results | | | |
| WBC (cells *10^9/l) | 4.7 (3.7–5.9) | 3.4 (2.6–4.8) | 3 (2.5–4.2) |
| RBC (cells *10^12/l) | 4.6 (4.1–4.9) | 4 (3.4–4.6) | 3.9 (2.8–4.3) |
| Hemoglobin (g/l) | 140 (129–154) | 123 (109–141) | 116 (78–131) |
| Platelets (cells *10^9/l) | 150 (116–181) | 70 (54–112) | 60 (52–84) |
| INR | 0.99 (0.92–1.06) | 1.09 (0.99–1.26) | 1.09 (1.02–1.24) |
| Prothrombin time (sec) | 11.5 (10.8–12.6) | 12.7 (11.5–14.5) | 12.7 (11.8–14.4) |
| ALT (IU/l) | 34 (20–122) | 37 (22–63) | 36 (22–45) |
| AST (IU/l) | 37 (27–91) | 45 (31–65) | 42 (29–60) |
| AKP (IU/l) | 85 (66–117) | 100 (73–133) | 82 (67–126) |
| GGT (IU/l) | 35 (17–82) | 42 (22–98) | 37 (20–65) |
| Total bilirubin (umol/l) | 20 (15–26) | 28 (21–43) | 26 (19–36) |
| Albumin (g/l) | 43 (38–46) | 38 (33–44) | 38 (34–43) |
| Creatinine (umol/l) | 65 (58–74) | 67 (55–76) | 65 (55–75) |
| Urea nitrogen (mmol/l) | 4.5 (3.8–5.3) | 4.7 (3.9–6.2) | 4.8 (3.9–7.4) |
| Blood glucose (mmol/l) | 5.3 (5–5.8) | 5.6 (4.9–6.6) | 5.8 (5.1–7.1) |
| MAP (mmHg) | 90 (83–97) | 92 (87–100) | 91 (87–102) |
| Derivation results | | | |
| MELD Score | 4.1 (1.9–6.3) | 6.7 (4.2–9) | 5.6 (3.4–7.8) |
| FIB-4 Index | 2.6 (1.6–3.9) | 5.7 (3.1–9.3) | 6.3 (4–8.4) |
| MRI results | | | |
| PSVR (n/mm^3)/cm^3 | 645 (420–1040) | 135 (78.8–232) | 118 (72.9–173) |
| PSDR (n/mm^3)/mm | 1170 (960–1530) | 480 (343–716) | 433 (333–570) |
| SV (cm^3) | 234 (173–309) | 514 (397–757) | 552 (438–928) |
| SD (cm) | 12.5 (10.6–14.3) | 15.3 (12.9–16.8) | 15.5 (12.6–16.8) |

Qualitative data were expressed as numbers and percentages (%); quantitative data were expressed as median (25%–75% quantiles).

Abbreviations: RBC, red blood cell; WBC, white blood cell; ALT, alanine aminotransferase; AST, aspartate aminotransferase; AKP, alkaline phosphatase; GGT, γ-glutamyl transpeptidase; INR, international normalized ratio; MELD, a model for end-stage liver disease; PSVR, PLT count/spleen volume; MAP, mean arterial pressure; PSDR, PLT count/spleen longest diameter of spleen; SV, Spleen volume; SD, Spleen longest diameter; FIB-4 Index = Year×AST/(PLT×ALT$^{0.5}$).

of Child-Pugh A patients, all five parameters (PSVR, PSDR, SV, SD, platelet count) were significantly different (p < 0.001) between the two groups of patients with EVs or HRVs.

The characteristics of EV and HRV of different non-invasive models in 199 patients and the Child-Pugh A group was shown in Table 2. The AUC of PSVR for identifying EV or HRV

**Table 2. The ability of each group to diagnose EV or HRV.**

| All, n = 199 | | CV | AUC (95%CI) | P-value (Comparison with PSVR's AUC) | P-value (Comparison with SV's AUC) | Acc | Sen | Spe | PPV | NPV |
|---|---|---|---|---|---|---|---|---|---|---|
| EV | PSVR | 346 | 0.907 (0.862–0.953) | - | - | 0.849 | 0.877 | 0.831 | 0.780 | 0.907 |
| | PSDR | 780 | 0.884 (0.834–0.934) | 0.035 | 0.603 | 0.834 | 0.827 | 0.839 | 0.779 | 0.876 |
| | SV | 343 | 0.895 (0.844–0.946) | 0.400 | - | 0.844 | 0.864 | 0.831 | 0.778 | 0.899 |
| | SD | 14.8 | 0.773 (0.707–0.839) | <0.001 | <0.001 | 0.739 | 0.580 | 0.847 | 0.723 | 0.746 |
| | PLT | 138 | 0.862 (0.808–0.917) | 0.012 | 0.260 | 0.774 | 0.926 | 0.669 | 0.658 | 0.929 |
| | PSDR909 | 909 | | - | - | 0.809 | 0.864 | 0.771 | 0.722 | 0.892 |
| HRV | PSVR | 242 | 0.855 (0.790–0.920) | - | - | 0.824 | 0.914 | 0.763 | 0.725 | 0.928 |
| | PSDR | 623 | 0.848 (0.786–0.910) | 0.606 | 0.400 | 0.809 | 0.852 | 0.780 | 0.726 | 0.885 |
| | SV | 408 | 0.821 (0.739–0.904) | 0.096 | - | 0.769 | 0.829 | 0.756 | 0.420 | 0.954 |
| | SD | 15.3 | 0.701 (0.607–0.796) | <0.001 | <0.001 | 0.769 | 0.571 | 0.811 | 0.392 | 0.899 |
| | PLT | 99 | 0.853 (0.797–0.910) | 0.941 | 0.427 | 0.769 | 0.827 | 0.729 | 0.677 | 0.860 |
| CP A group, n = 147 | | | | | | | | | | |
| EV | PSVR | 346 | 0.902 (0.843–0.962) | - | - | 0.849 | 0.852 | 0.847 | 0.793 | 0.893 |
| | PSDR | 880 | 0.879 (0.813–0.944) | 0.085 | 0.739 | 0.824 | 0.827 | 0.822 | 0.761 | 0.874 |
| | SV | 347 | 0.889 (0.822–0.955) | 0.488 | - | 0.844 | 0.833 | 0.848 | 0.727 | 0.913 |
| | SD | 14.8 | 0.786 (0.704–0.868) | <0.001 | <0.001 | 0.769 | 0.604 | 0.848 | 0.659 | 0.816 |
| | PLT | 138 | 0.855 (0.783–0.927) | 0.026 | 0.348 | 0.779 | 0.914 | 0.686 | 0.667 | 0.920 |
| | PSDR909 | 909 | | - | - | 0.81 | 0.833 | 0.798 | 0.667 | 0.908 |
| HRV | PSVR | 206 | 0.926 (0.884–0.969) | - | - | 0.894 | 0.951 | 0.856 | 0.819 | 0.962 |
| | PSDR | 761 | 0.917 (0.872–0.962) | 0.464 | 0.442 | 0.869 | 1 | 0.780 | 0.757 | 1 |
| | SV | 409 | 0.885 (0.802–0.968) | 0.164 | - | 0.823 | 0.905 | 0.810 | 0.442 | 0.981 |
| | SD | 15.0 | 0.775 (0.668–0.882) | 0.001 | 0.014 | 0.776 | 0.714 | 0.786 | 0.357 | 0.943 |
| | PLT | 83 | 0.915 (0.864–0.965) | 0.561 | 0.530 | 0.849 | 0.852 | 0.847 | 0.793 | 0.893 |

1. SV, SD had changed the state variable. 2. Abbreviations: CV, Cutoff value; AUC, the area under the ROC curve; Acc, Accuracy; Sen, Sensitivity; Spe, Specificity; PPV, Positive predictive value; NPV, Negative predictive value.

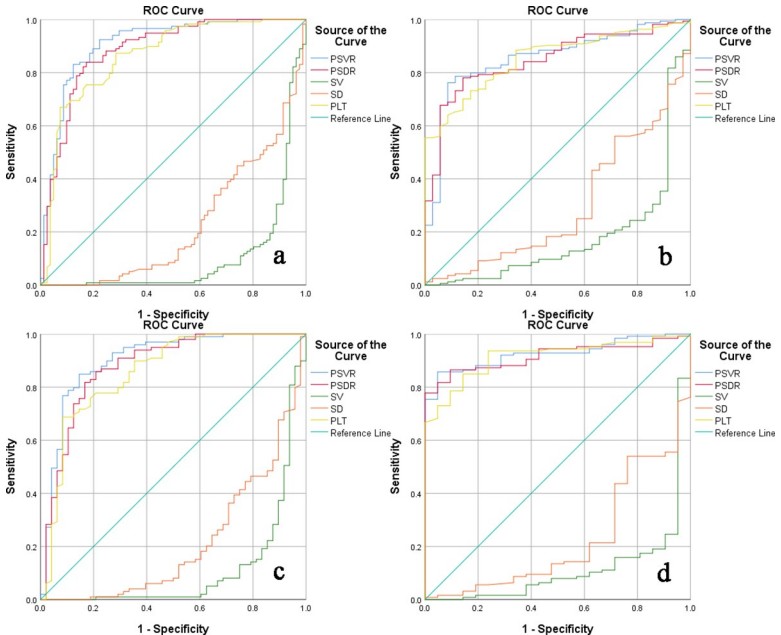

**Fig 2. The ROC curve of relevant indicators in the diagnosis of esophageal varices.** a. ROC curve for diagnosis of EV in all populations; b. ROC curve for diagnosis of HRV in all populations; c. ROC curve for diagnosis of EV in CP A patients; d. ROC curve for diagnosis of HRV in CP A patients.

in each group (85.5%-92.6%) was higher than the other four parameters (Fig 2), however, the diagnostic performance of PSVR was superior to that of PSDR only in identifying EV (P = 0.035). The AUC of PSDR for diagnosing HRV was higher than SV, and the AUC of SV for diagnosing EV was higher than PSDR, but the difference was not significant (P>0.05). The values were also calculated for the platelet count/spleen diameter ratio cut-off of 909 (obtained in the study by Giannini et al.), the sensitivity and specificity obtained in our study were 86.4% and 77.1% respectively in 199 patients, which were lower than those of the original study by Giannini et al. [11, 12].

The P-value, coefficient of regression, and OR value of related variables for EV, HRV are shown in Table 3. In Child-Pugh A patients, multivariate logistic regression analysis showed PSVR could be a predictor of HRV (P<0.05), SV was a reliable predictor of EV (P<0.05). However, large samples and multi-center clinical studies are needed.

**Table 3. P-value and OR value of related variables for EV, HRV in CP A patients.**

|  | EV | | | HRV | | |
|---|---|---|---|---|---|---|
|  | RC | OR (95% CI) | P value | RC | OR (95% CI) | P value |
| PSVR |  |  | 0.123 | -0.114 | 0.892 (0.804–0.990) | 0.032 |
| PSDR |  |  | 0.915 | -0.061 | 0.941 (0.890–0.995) | 0.033 |
| SV | 0.070 | 1.072 (1.024–1.123) | 0.003 |  |  | 0.151 |
| SD | 0.254 | 1.289 (1.008–1.649) | 0.043 |  |  | 0.709 |
| PLT |  |  | 0.219 | -0.058 | 0.943 (0.890–0.999) | 0.047 |

1. Obtained by binary logistic regression, variables considered MELD, prothrombin time, total bilirubin, albumin, ascites, RPR, leukocytes, hemoglobin, urea nitrogen, albumin/globulin, total bile acid. To increase the regression coefficients, PSVR, PSDR, and SV were reduced to 10 times the original size.

2. RPR = Erythrocyte volume distribution width / PLT; Abbreviations: RC, Regression coefficient.

## Discussion

The PSDR diagnostic procedure was first proposed in 2003. To date, it remains an essential assessment tool due to its perception as an effective EV prediction method in patients with cirrhosis of the liver. However, there are also many disagreements regarding its role in predicting EV due to the inconsistent etiology and degree of cirrhosis in numerous study subjects. Previously, spleen diameter was typically measured by ultrasound. In this study, spleen volume was measured by simulating a standard long ellipse, after which the longest of the three diameters was selected as the spleen diameter, a more accurate method of finding the bipolar diameter. In addition, most of the current spleen volume measurements are derived from calculating the splenic bipolar diameter, multiplied by the thickness, which will result in a disproportionately high value being measured. Compared with spleen diameter, spleen volume has a higher rate of diagnosis, regardless of EV or HRV; in the absence of serological indicators, this result may be due to spleen volume's more accurate reflection of the degree of splenomegaly than spleen diameter resulting in higher accuracy.

In our study, when using 909 as the cutoff value, the positive and negative predictive values of the PSDR diagnosis of EV were 73% and 88%, respectively, which differed from Giannini et al.'s 74% and 100% EV values. This difference may be due to distinct genetic characteristics within the study's European and American source populations. A meta-analysis [13] of 49 related studies showed that the AUC of PSDR for the diagnosis of GEV and HRV was 0.872 and 0.813, respectively. Here, the AUC of PSDR for EV was higher than that of HRV, which is not very different from our findings. Another META analysis [14] of 71 related studies demonstrated PSDR's improved accuracy in diagnosing GEV and HRV compared with spleen diameter or platelet count. This finding indicated that the serological indicators were not as reliable as the combined indicators. Likely, these parameters may vary with blood draw time, sample transport time, and liver function status.

Comparing PSDR, SV, spleen diameter, and platelet count, the PSVR showed the highest accuracy (82.4%-89.4%) for diagnosing EV or HRV in each group. Meanwhile, in the diagnosis of HRV, the AUC of PSDR was higher than SV, and the AUC of SV was more elevated than PSDR in the diagnosis of EV; however, the difference was not statistically significant. This trend demonstrates the possible use of SV instead of PSDR in predicting EV or HRV in the absence of serological examinations.

Since the main etiology of cirrhosis in China is hepatitis B viral infection, this study only considered patients with cirrhosis caused by hepatitis B. We excluded other causes and cases where cirrhosis was caused by hepatitis B in conjunction with other triggers. Hepatitis B liver cirrhosis has a more extended compensation period; however, during the decompensation period, various complications, including gastroesophageal variceal bleeding, portal hypertensive gastropathy, ascites, and jaundice, reduce patient survival rates. In contrast to patients with decompensated cirrhosis, patients in Child-Pugh A require more gastroscopy to clarify the presence or absence of complications such as EV, compared with painless gastroscopy, ordinary gastroscopy procedures have fewer contraindications and are significantly cheaper. In most hospitals in China, MRIs are slightly more expensive than ordinary gastroscopies, but they can diagnose liver and spleen disease while also more accurately predicting esophageal varices and their degree, and have greater clinical diagnostic significance in patients with contraindications to gastroscopy or unwillingness to undergo gastroscopy. If unnecessary gastroscopies are performed, complications such as tooth damage, perforation, and bleeding may occur [15]. Thus, PSVR and SV eliminate this risk by facilitating the exemption of 67% of Child-Pugh A patients from the gastroscopy procedure. For HRV, SV showed the highest negative predictive value (95.4%), and the missed diagnosis rate (8.6%) of PSVR in the total

patients was the lowest, suggesting that high-risk esophageal varices can be safely excluded for PSVR and SV, which were valuable in diagnosing EV or HRV. For non-invasive testing to be practical, it must have a high negative predictive value. Thus, the missed EV diagnosis in patients with liver cirrhosis is a significant risk and has severe harmful consequences.

Multiple external factors may have affected the experimental outcome. For instance, 55 patients (27.6%) in the study had a history of drinking; some studies have even shown that ethanol can shorten the life span of platelets [16], leading to their decrease. However, the evidence for the diagnosis of alcoholic cirrhosis in this study was insufficient; MRI determination of spleen volume requires specific training for accurate measurements. Furthermore, while the length and volume of the spleen were significantly and independently associated with sex, height, and weight [17], we did not explore this possible correlation; this requires more in-depth, targeted research. Also, as the spleen is not a standard oblong ellipsoid, and there may be errors in the obtained values; not all liver cirrhosis diagnoses were based on liver biopsy, as some were based on clinical non-invasive results. In addition, since the management of varices depends not only on whether they are high-risk or not, but also on the size of the varices, studies related to large varices are yet to be performed to clarify the predictive power of PSVR, spleen volume.

MRIs combined with laboratory tests can more effectively assess the possibility of severe complications in high-risk patients. This diagnostic method facilitates timely patient interventions, producing better specificity in low-risk groups, and further reducing examination costs. Studies have shown that patients without varices may show a decrease in PSDR and are more likely to develop EV [18]. Similarly, we confirmed the effectiveness of PSVR compared to the original PSDR in diagnosing EV, but it is not statistically significant in predicting EV in Child-Pugh A patients compared to PSDR (P>0.05), which requires further study. Furthermore, SV can be used instead of PSDR for EV or HRV prediction when there is no serological examination, and dynamic short-term monitoring of these two parameters will help determine whether a patient (especially Child-Pugh A patients) should undergo gastroscopies. Therefore, it may be clinically possible to use PSVR or spleen volume as part of the diagnostic workup for patients with hepatitis B cirrhosis, but further multicenter, prospective, and large sample studies are needed.

## Supporting information

**S1 Checklist. STARD checklist for reporting of studies of diagnostic accuracy.**
(DOC)

**S1 Database.**
(SAV)

**S1 Graphical abstract.**
(TIF)

## Acknowledgments

The authors would like to thank all the staff of the Department of Radiology, Gastroscopy Room, and Gastroenterology Department of Xiangyang No.1 People's Hospital for their support.

## Author Contributions

**Conceptualization:** Sihao Yu, Wei Chen, Zicheng Jiang.

**Data curation:** Sihao Yu, Wei Chen, Zicheng Jiang.

**Formal analysis:** Sihao Yu.

**Investigation:** Sihao Yu, Wei Chen, Zicheng Jiang.

**Project administration:** Wei Chen, Zicheng Jiang.

**Resources:** Sihao Yu, Zicheng Jiang.

**Software:** Sihao Yu.

**Supervision:** Wei Chen, Zicheng Jiang.

**Validation:** Wei Chen, Zicheng Jiang.

**Visualization:** Sihao Yu.

**Writing – original draft:** Sihao Yu.

**Writing – review & editing:** Sihao Yu, Wei Chen, Zicheng Jiang.

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
