## [Decision Letter · Decision Letter 0]

9 Nov 2021

PONE-D-21-33951Platelet count/spleen volume ratio has a good predictive value for esophageal varices in patients with hepatitis B liver cirrhosisPLOS ONE

Dear Dr. Yu,

Thank you for submitting your manuscript to PLOS ONE. After careful consideration, we feel that it has merit but does not fully meet PLOS ONE’s publication criteria as it currently stands. Therefore, we invite you to submit a revised version of the manuscript that addresses the points raised during the review process.

We look forward to receiving your revised manuscript.

Kind regards,

Gopal Krishna Dhali, MBBS, MD, DM

Academic Editor

PLOS ONE

Journal Requirements:

Additional Editor Comments:

Kindly correct the numerical error on page 9, line 166 and page 12, line 229.

Reviewers' comments:

Reviewer's Responses to Questions

**Comments to the Author**

1. Is the manuscript technically sound, and do the data support the conclusions?

Reviewer #1: Yes

Reviewer #2: Yes

2. Has the statistical analysis been performed appropriately and rigorously? 

Reviewer #1: Yes

Reviewer #2: Yes

3. Have the authors made all data underlying the findings in their manuscript fully available?

Reviewer #1: Yes

Reviewer #2: Yes

4. Is the manuscript presented in an intelligible fashion and written in standard English?

Reviewer #1: Yes

Reviewer #2: Yes

5. Review Comments to the Author

Reviewer #1: Well conceptualized study. Thanks to the authors for picking up this particular issue for their investigation. Statistical analyses were performed adequately as was suggested. Manuscript contains all the data related to the study observations. Authors complied with the suggestions provided in the previous review of the manuscript (ID: PONE-D-21-30287). I would request authors to continue this study with the compensated cirrhosis only.

Reviewer #2: 1. Well thought concept.

2. English grammar can be improved for the entire article.

3. Following correction to be made.

(See line 158, page 9): Mean age and range/IQR is reported. It should be median and range/IQR.

6. PLOS authors have the option to publish the peer review history of their article (what does this mean?). If published, this will include your full peer review and any attached files.

Reviewer #1: **Yes: **KAUSIK DAS, Dept of Hepatology, IPGME&R, Kolkata, India

Reviewer #2: **Yes: **Debashis Misra

---

## [Author Response · Author response to Decision Letter 0]

13 Nov 2021

Dear Editors and Reviewers:

Thank you for your letter dated November 10. We were pleased to know that our work was rated as potentially acceptable for publication in Journal, subject to minor revision. We thank the editors and reviewers for the time and effort that they have put into reviewing the previous version of the manuscript. Their suggestions have enabled us to improve our work. Based on the instructions provided in your letter, we uploaded the file of the revised manuscript. Accordingly, we have uploaded a copy of the original manuscript with all the changes highlighted in MS Word. Appended to this letter is our point-by-point response to the comments raised by the reviewers. The comments are reproduced and our responses are given directly afterward in a different color (red). We would like also to thank you for allowing us to resubmit a revised copy of the manuscript.

We hope that the revised manuscript is accepted for publication in the Journal of PLOS ONE.

The modified parts are marked in red in the copy paper. The main corrections in the paper and the responds to the academic editor and reviewers’ comments are as following: 

Responds to the academic editor’s comments: 

Since this is a retrospective study and the methods are written in detail in the Materials and methods and Statistical analysis sections, there is no need to store it in protocols.io.

We have carefully read the documents PLOSOne_formatting_sample_main_body.pdf and PLOSOne_formatting_sample_title_authors_affiliations.pdf for PLOS ONE's style requirements, file naming requirements and made changes accordingly.

 We reviewed the references again to ensure that they were complete and correct.

Additional Editor Comments:

Kindly correct the numerical error on page 9, line 166 and page 12, line 229.

 We have corrected 57% to 67% (page 9, line 162 and page 21, line 276). We are very sorry for the trouble we have caused you.

 Thank you very much for your dedication!

Responds to the reviewer’s comments: 

Reviewer #1: Well conceptualized study. Thanks to the authors for picking up this particular issue for their investigation. Statistical analyses were performed adequately as was suggested. Manuscript contains all the data related to the study observations. Authors complied with the suggestions provided in the previous review of the manuscript (ID: PONE-D-21-30287). I would request authors to continue this study with the compensated cirrhosis only.

Response to comment: Thank you for your compliments and we will continue to look for more meaningful inspiration. We will consider the next study in a population with compensated cirrhosis. Thank you very much again for your advice!

Reviewer #2: 1. Well thought concept.

Response to comment: Thank you for your praise and we will continue to work on more creative attempts.

2. English grammar can be improved for the entire article.

Response to comment: Our manuscript has been corrected in English grammar by 3 people since the beginning, but it still doesn't work very well, if possible, can you tell us where the problem is very big? Thank you very much for your advice.

3. Following correction to be made.

(See line 158, page 9): Mean age and range/IQR is reported. It should be median and range/IQR.

Response to comment: We are very sorry for our repeated mistakes and have corrected the phrase. (See line 154, page 9)

Thank you very much for your comments, we will work harder to learn English to make the language more vivid.

 We sincerely thank the editors and reviewers for their enthusiastic work and hope that the corrections will be accepted. 

 Again, thank you so much for your comments and suggestions!

Best regards

Sincerely yours

Sihao Yu

(On behalf of co-authors)

Detailed address

Tel: (+86)15997812001 Email: 1034137401@qq.com

---

## [Editor Report · Decision Letter 1]

17 Nov 2021

Platelet count/spleen volume ratio has a good predictive value for esophageal varices in patients with hepatitis B liver cirrhosis

PONE-D-21-33951R1

Dear Dr. Yu,

We’re pleased to inform you that your manuscript has been judged scientifically suitable for publication and will be formally accepted for publication once it meets all outstanding technical requirements.

Kind regards,

Gopal Krishna Dhali, MBBS, MD, DM

Academic Editor

PLOS ONE

---

## [Editor Report · Acceptance letter]

19 Nov 2021

PONE-D-21-33951R1 

Platelet count/spleen volume ratio has a good predictive value for esophageal varices in patients with hepatitis B liver cirrhosis 

Dear Dr. Yu:

I'm pleased to inform you that your manuscript has been deemed suitable for publication in PLOS ONE. Congratulations! Your manuscript is now with our production department. 

Kind regards, 

on behalf of

Dr. Gopal Krishna Dhali 

Academic Editor

PLOS ONE